# Diagnosis and Management of Non-Variceal Gastrointestinal Hemorrhage: A Review of Current Guidelines and Future Perspectives

**DOI:** 10.3390/jcm9020402

**Published:** 2020-02-02

**Authors:** Sobia Mujtaba, Saurabh Chawla, Julia Fayez Massaad

**Affiliations:** Division of Digestive Diseases, Emory University, 1365 Clifton Road, Northeast, Building B, Suite 1200, Atlanta, GA 30322, USA; sobia.mujtaba@emory.edu (S.M.); saurabh.chawla@emory.edu (S.C.)

**Keywords:** gastrointestinal bleeding, risk factors, non-steroidal anti-inflammatory drugs, endoscopy, guidelines, hemostasis

## Abstract

Non-variceal gastrointestinal bleeding (GIB) is a significant cause of mortality and morbidity worldwide which is encountered in the ambulatory and hospital settings. Hemorrhage form the gastrointestinal (GI) tract is categorized as upper GIB, small bowel bleeding (also formerly referred to as obscure GIB) or lower GIB. Although the etiologies of GIB are variable, a strong, consistent risk factor is use of non-steroidal anti-inflammatory drugs. Advances in the endoscopic diagnosis and treatment of GIB have led to improved outcomes. We present an updated review of the current practices regarding the diagnosis and management of non-variceal GIB, and possible future directions.

## 1. Introduction

Non-variceal gastrointestinal bleeding (GIB) is a common problem worldwide that is encountered in the ambulatory and hospital settings and is associated with significant morbidity and mortality. The annual rate of hospitalization for any type of GIB in the United States (U.S) is estimated to be 350 hospital admissions/100,000 population, with more than 1,000,000 hospitalizations every year, and a mortality rate of 2–10% [1,2,3]. Hemorrhage form the gastrointestinal (GI) tract is categorized as upper GI bleeding (UGIB), small bowel bleeding (also formerly referred to as obscure GIB (OGIB)) or lower GIB (LGIB). Among hospitalized patients with GIB, 40% are attributed to UGIB, 25% LGIB and 35% in an undefined location [4].

Herein we present a brief review on the etiology, manifestations and recommended guidelines on the management of GIB, with a focus on endoscopic diagnosis and management.

### 1.1. Initial Assessment

The initial evaluation of any patient presenting with GIB, regardless of etiology rests on adequate resuscitation, obtaining a history, physical examination and laboratory tests that can guide decisions regarding triage, empiric medical therapy, and further diagnostic testing or therapeutic interventions. Another important aspect of the initial assessment includes a careful review of medications of a patient presenting with a GIB, with particular attention to use of antiplatelet agents and anticoagulants. The management of antiplatelet agents and anticoagulants is beyond the scope of this discussion, and is addressed in details elsewhere in the literature [5]. Briefly, where feasible these medications must be held and coagulopathy should be corrected to allow endoscopic interventions to be carried out safely.

### 1.2. Bleeding Manifestations

GIB that is clinically apparent as visible blood loss may manifest as hematemesis, melena or hematochezia. Hematemesis includes or refers to vomiting of either red blood or coffee-ground emesis, and suggests bleeding proximal to the ligament of Treitz. Melena is defined as black, tarry stool that occurs several hours after the bleeding event and results from the degradation of blood to hematin or other hemochromes by gut bacteria and can be seen with variable degrees of blood loss, being visible with as little as 50 mL of blood [6,7]. Hematochezia refers to red or maroon blood in the stool, and suggests active bleeding usually due to LGIB, but can been in massive UGIB, typically associated with hemodynamic instability [8].

OGIB [9] is defined as either overt or occult bleeding of unknown origin that recurs or persists after an initial endoscopic evaluation including esophagogastroduodenoscopy (EGD) and ileocolonoscopy. Obscure overt GIB is clinically apparent, whereas occult bleeding refers to OGIB that is not apparent and usually presents as unexplained iron deficiency anemia (IDA) or a positive fecal occult blood test.

### 1.3. Brief Overview of Common Endoscopic Hemostatic Devices

Once a diagnosis of GIB is established, therapeutic hemostasis can be achieved endoscopically via injection therapy, ablative methods, and by mechanical therapy depending upon the lesion [10]. The application of these devices id discussed in further detail in each section with regards to specific lesions, a brief overview is provided below.

The mechanism of hemostasis of injection therapy with saline or dilute epinephrine (diluted with saline to 1:10,000 to 1:20,000) is local tamponade [10,11]. Dilute epinephrine injected into four quadrants within 3 mm of the bleeding site also results in local vasospasm [11,12,13] and allows for a cleaner field of vision for targeted treatment of the bleeding site. Sclerosants (e.g., absolute alcohol, sodium tetradecyl sulfate, polidocanol, and ethanolamine) are tissue irritants that cause endofibrosis and vascular obliteration through vascular thrombosis and endothelial damage when injected into or adjacent to blood vessels, but must be used cautiously in small amounts due to concern for tissue necrosis [14,15,16].

Thermal coagulation can be achieved by contact probes or argon plasma coagulation (APC). Contact probes achieve hemostasis by sealing (coaptation) of the underlying vessel which involves simultaneous application of pressure to the vessel with the probe to compress it while coagulation is performed [17]. APC is a non-contact thermal method of hemostasis that uses ionized non-flammable argon gas to deliver thermal energy that results in coagulation of the target tissue. The power settings for the APC probes differ by regions of the GI tract, with higher settings used for thicker tissue such as the stomach and lower settings for thinner tissue such as the small bowel and colon.

The mechanism of hemostasis with application of hemoclips is synonymous to that of surgical ligation by grasping vessels or surrounding tissue. They do not cause tissue injury or inflammation which is seen with sclerosants or thermal coagulation [18]. With regard to peptic ulcer disease (PUD) related bleeding, clips may be advantageous in that they can be used for retreatment of recurrent bleeding after initial hemostasis with thermal therapy. Evidence from animal studies suggests that ulcer healing may be accelerated when sides are opposed by placing the endoclip across the ulcer into normal appearing tissue when compared to thermal therapy [19].

TC-325 is a novel endoscopic hemostatic powder/spray that is safe and effective for treatment of UGIB and LGIB. Hemospray is a highly absorptive inorganic mineral nanopowder that can be sprayed onto the surface of actively bleeding lesions forming a mechanical barrier which can lead to immediate hemostasis and is shown to promote clot formation and shorten coagulation time [20]. The success rate with nanopowder for non-variceal UGIB (including ulcers and other lesions) varies between 75% to 100%, with rebleeding rates between 10% to 49% [21,22,23,24,25,26,27,28,29,30]. Hemospray is also effective for hemostasis of LGIB [29,30,31]. Hemospray may be used in patients with malignant GI bleeding that may be difficult to control with standard endoscopic techniques [32]. However, the time during which the powder remains in the GI tract is short and is eliminated as early as within 24 h of application.

## 2. Upper Gastrointestinal Bleeding

UGIB refers to GIB originating proximal to the ligament of Treitz [33]. It can manifest as hematemesis, coffee ground emesis and/or melena. Hematochezia may be seen in 5–10% of patients with a briskly bleeding upper GI source [34,35].

UGIB represents a substantial clinical burden. It accounts for more than 300,000 admissions annually with an estimated mortality of 3.5% to 10% [36,37], with a higher incidence among men and the elderly [33,38]. The incidence of rebleeding in UGIB is estimated to be 5% to more than 20%, and is associated with a higher risk of mortality [38,39]. However, some data suggests that the incidence of UGIB may be declining [36,38,40] in tandem with a declining incidence of PUD [3,41].

The most common etiologies of non-variceal UGIB in descending order of frequency are PUD (20–50%), gastroduodenal erosions (8–15%), esophagitis (5–15%), Mallory Weiss syndrome (8–15%), vascular malformations (5%), and other less common causes (malignancy, Dieulafoy’s lesions, gastric antral vascular ectasia (GAVE), hemobilia, hemosuccus pancreaticus, aortoenteric fistula, Cameron lesions) [37,42,43,44,45,46,47].

The four major risk factors for bleeding gastroduodenal ulcers are Helicobacter Pylori (H.Pylori) infection, nonsteroidal anti-inflammatory drug (NSAID) use, stress and gastric acid secretion [48,49]. The clinical course of PUD may differ by the underlying etiology [50,51]. In a large study of 2242 patients with UGIB, of the 575 (26%) who had PUD, those with H.Pylori associated ulcers had the lowest rates of rebleeding and mortality, H.Pylori-negative ulcers were associated with poorer outcomes regardless of NSAID use and, those with H.Pylori negative ulcers and no NSAID use had the worst outcomes and had more severe systemic disease [52].

### 2.1. Risk Stratification

Early risk stratification is recommended in patients presenting with acute UGIB to identify patients at higher risk of bleeding or death and guide management decisions regarding timing of endoscopy, triage to appropriate level of care, and disposition [18,53,54]. Commonly used, validated risk stratification tools include the Glasgow Blatchford score (GBS), [55] the Clinical Rockall score, [56] and the AIMS65 score [57]. The GBS (scored 0–23) [55] can be used to perform pre-endoscopy risk stratification using parameters of systolic blood pressure (sbp), hemoglobin concentration, blood urea nitrogen, melena, syncope and the presence of heart disease or heart failure. The score is more sensitive than the Rockall score [56] in identifying high-risk patients [58] and in identifying many low-risk patients who can be managed as outpatients [58,59,60]. The Rockall score [56] can be calculated before endoscopy (scored 0–7) or after endoscopy and is superior to the GBS in predicting patient mortality [61,62,63]. The AIMS65 [57] is a newer risk stratification score that uses patients factors (age, sbp, mental status) and lab values (INR and albumin) and is highly accurate for predicting in-house hospital mortality among patients with UGIB [57,64,65].

### 2.2. Medical Management

The use of intravenous (IV) proton pump inhibitors (PPI) is recommended in patients with suspected UGIB [18,66]. Evidence from in vitro studies suggests that an intra-gastric pH >6 promotes clot formation and stability [67,68], and a strongly acidic intra-gastric milieu leads to an inhibition of platelet aggregation, plasma coagulation and promotes lysis of formed clots [69]. In a landmark trial in 2007, investigators in Hong Kong randomized patients with suspected UGIB to pre-endoscopic IV PPI therapy or placebo. The group receiving PPI therapy had less active bleeding, less need for therapy during endoscopy but there were no differences in transfusion requirements, length of stay, need for surgery or mortality [70]. In patients with UGIB secondary to PUD, pre-endoscopic IV PPI therapy reduces the proportion of high-risk lesions (active bleeding, non-bleeding visible vessel, and adherent clot) at index endoscopy, and the need for endoscopic therapy, however it is not associated with better outcomes of mortality, need for surgery or rebleeding [71]. In those cases where endoscopy is delayed or cannot be performed, IV PPI therapy is associated with reduced rate of recurrent bleeding and progression to surgery, but not mortality [72]. Intermittent PPI dosing is considered non-inferior to continuous therapy with regard to rebleeding, need for surgery or repeat intervention and mortality [73].

### 2.3. Prokinetics

The use of prokinetic agents should be considered in cases where there is a high probability of encountering fresh blood or clots in the stomach to improve gastric visualization, and diagnostic yield during endoscopy [18,66]. Infusions of erythromycin or metoclopramide 20–120 min prior to endoscopy in patients with acute UGIB are associated with improved visualization of mucosa, decreased need for second look endoscopy to determine site and cause of bleeding, but are not proven to show any benefit with regard to transfusion requirements, duration of hospital stay or surgery [74].

### 2.4. Role of Endoscopy in Management of Upper Gastrointestinal Bleeding

Endoscopic evaluation, particularly early endoscopy within 24 h of presentation is associated with decreased transfusion requirements and length of hospitalization [53,54,75,76]. Therefore, early endoscopy is recommended in patients presenting with a non-variceal UGIB [53,54,66]. Endoscopy can also identify patients with low-risk lesions who can be managed as outpatients [54,77,78].

### 2.5. Endoscopic Therapy for Bleeding Peptic Ulcer Disease

Endoscopic therapy is indicated for the treatment of ulcers with stigmata of recent hemorrhage (SRH) that carry a higher risk for recurrent bleeding and mortality [18,66]. These stigmata as described by the Forrest classification [79] in descending order of risk of rebleeding are: Class Ia, spurting hemorrhage (90%); Class IIa, nonbleeding visible vessel (50%) (Figure 1), Class Ib, oozing hemorrhage (10–20%); Class IIb, adherent clot (25–30%) (Figure 2); Class IIc, flat pigmented spot (7–10%) and; Class III, clean ulcer base (3–5%). Most patients with ulcer bleeding have low risk Forrest IIc and Forrest III lesions [45], and endoscopic treatment is not recommended [18,66].

Endoscopic treatment is recommended for actively bleeding ulcers, and for non-bleeding visible vessel [18,66]. Ulcers with overlying clots should be irrigated to assess and treat the underlying lesion if feasible. The management of adherent clots that cannot be removed by irrigation or gentle suction is controversial with conflicting results noted in two meta-analyses of randomized controlled trials (RCTs) with regard to risk of rebleeding, need for surgery, and mortality [15]. There are guidelines that recommend application of endoscopy combined with injection treatment and thermal coagulation or mechanical device along with a PPI [54], while others suggest that intensive PPI therapy alone may also be sufficient in these cases [53]. In the absence of consensus guidelines, the decision to treat endoscopically in patients with adherent clots should be made on an individual basis taking into consideration patient characteristics that may be associated with a higher risk of bleeding (e.g., age, co-morbid conditions) [18,66].

Endoscopic interventions that are singularly effective and non-inferior to each other in achieving hemostasis in bleeding ulcers include injection therapy with agents other than epinephrine e.g., sclerosants (e.g., absolute alcohol, polidocanol, and ethanolamine), thermal ablation (bipolar or monopolar electrocoagulation, APC, heater probe, and laser), thrombin/fibrin glue, and hemoclips [15]. However, in several meta-analyses, combination therapy with epinephrine injection and cautery or hemoclips was found to be superior to epinephrine alone and reduced the risk of recurrent bleeding, progression to surgery, and mortality [15,80,81,82]. Therefore, the standard recommended approach is combination therapy with epinephrine and either coaptive thermal therapy or hemoclip placement or thrombin/fibrin glue [18,66]. Hemospray may be considered as a temporizing measure in active bleeding when endoscopic therapy fails [53]. Following endoscopic therapy of ulcers with high risk SRH, IV PPI therapy for 72 h effectively reduces the rate of recurrent bleeding and mortality [15,72,83,84].

### 2.6. Endoscopic Therapy for Other Non-Variceal Causes of Upper GI Bleeding

Mallory–Weiss tears are characterized by longitudinal mucosal lacerations at the gastroesophageal junction, gastric cardia or distal esophagus, which are usually associated with forceful retching [85]. Bleeding from submucosal arteries is typically self-limited [86] but ongoing or severe bleeding requires endoscopic therapy such as epinephrine injection, thermal coagulation, clip placement, and band ligation [87,88]. Bleeding that is refractory to endoscopic therapy may require surgery or angiographic embolization [88].

Angiodysplasias or arteriovenous malformations (AVMs) are aberrant blood vessels that can be found throughout the GI tract and account for 4% to 7% of patients with UGIB [89,90]. They can occur sporadically or in association with chronic diseases e.g., end stage renal disease (ESRD), cirrhosis, collagen vascular diseases, hereditary hemorrhagic telengectasias, aortic stenosis and von Willebrand disease. Bleeding from AVMs often manifests as chronic blood loss with anemia but may present with overt bleeding. Effective hemostatic therapies include APC, bipolar or heater probed coagulation, ligation, and sclerotherapy [91,92].

Dieulafoy lesion (Figure 3), a dilated aberrant submucosal artery that erodes the overlying epithelium and then ruptures [43], is usually located in the proximal stomach along the lesser curvature, in close proximity to the GEJ but can be found anywhere in the GI tract [43,93]. The usual clinical manifestation is severe, intermittent bleeding. Endoscopic banding, clipping, electrocautery, laser therapy, heater probe, injection of sclerosants or epinephrine are all effective [66], however the use of epinephrine alone is associated with a higher risk of rebleeding [94] and band ligation may be associated with a higher risk of perforation in the small bowel and proximal colon [95].

GAVE or “water melon stomach” is seen in older women, in patients with cirrhosis and systemic sclerosis [96], and typically manifests as chronic blood loss with anemia. It is most commonly treated with APC, and more recently radiofrequency ablation (RFA), other options include endoscopic coagulation with a heater probe, bipolar probe, or laser therapy [97,98,99,100].

Gastrointestinal tumors, benign and malignant, account for less than 3% of cases of UGIB [101]. Neoplasms can cause bleeding from diffuse mucosal ulceration or from erosion into an underlying vessel. Standard endoscopic treatment modalities include injection therapy, thermal contact probes, and APC [102]. Hemostatic powder is effective for acute control of active bleeding, [32,103,104,105] however rebleeding in malignant lesions is common.

Endoscopic therapy can effectively control bleeding in the vast majority of patients with UGIB, but persistent or recurrent bleeding may be seen in 7–24% of cases [66,106]. Repeat endoscopic therapy is recommended [66] and is usually successful [107,108]. However, if bleeding cannot be controlled endoscopically, angiographic control should be considered [109,110].

### 2.7. Rare Causes of UGIB

Aorto-enteric fistulas, primary [111,112], or secondary [113], are most commonly located in the third or fourth portion of the duodenum, followed by the jejunum and ileum [114,115]. CT or CT angiography (CTA) are recommended as the first line diagnostic tests [66,116,117]. Endoscopy, preferably push enteroscopy (PE), may be required to exclude other causes of UGIB. Surgical repair is indicated in all cases.

Hemobilia, or bleeding from the hepatobiliary tract may be secondary to hepato-biliary diseases or injury, or from iatrogenic trauma from hepatic biliary tract instrumentation [118]. CT, magnetic resonance cholangiopancreatography (MRCP), or an endoscopic retrograde cholangiopancreatography (ERCP) can be useful for diagnosis. Selective hepatic artery angiography may reveal the source of bleeding [118,119]. Treatment involves either arterial embolization or surgery [118].

Hemosuccus pancreaticus, or bleeding from the pancreatic duct is most often found in patients with chronic pancreatitis, pancreatic pseudocysts, or pancreatic tumors and is confirmed through CT or MRCP, although a diagnostic ERCP may be considered [120,121,122]. Mesenteric artery embolization is the preferred first line treatment [120,122], surgery is indicated in refractory cases [123,124].

## 3. Small Bowel Bleeding

Small bowel bleeding previously referred to as OGIB, accounts for 5–10% of all sources of GI hemorrhage [125,126]. The term OGIB is applicable to those cases where the source of overt or occult bleeding anywhere in the GI tract cannot be identified with traditional bidirectional endoscopy, video capsule endoscopy (VCE), and device assisted enteroscopy (DAE) [127].

The most common etiologies of small bowel bleeding include inflammatory bowel disease (IBD), a Meckel’s diverticulum, Dieulafoy lesion, or a small bowel neoplasm (for e.g., GI stromal tumor, lymphoma, adenocarcinoma, carcinoid or polyp) in patients less than 40 years of age, and vascular lesions including angioectasias, ulcers or erosions secondary to NSAID use, and less commonly neoplasms in those over 40 years of age [128,129]. The evaluation of suspected small bowel bleeding typically starts with a repeating an upper endoscopic evaluation with either an EGD or PE and colonoscopy prior to evaluation of the small bowel with VCE, DAE, multiphase CT enterography (CTE), magnetic resonance enterography (MRE) or intraoperative enteroscopy (IOE). Definitive treatment of lesions that are amenable to endoscopic interventions, whether through PE or DAE, is similar to that in other parts of the GI tract. A suggested algorithm for endoscopic evaluation of suspected overt and occult small bowel bleeding by the American Society of Gastrointestinal Endoscopy (ASGE) is presented in Figure 4 [9].

### 3.1. Push Enteroscopy

PE is a deeper upper endoscopy (approximately 70 cm past the ligament of Treitz) performed using pediatric colonoscopy or a commercially available push enteroscope [127], allows for both diagnostic and therapeutic intervention. It is useful for examination of the distal duodenum and proximal jejunum. The yield of PE for a small bowel bleeding source is 24–70% [9,127].

### 3.2. Video Capsule Endoscopy

VCE is considered the test of choice in suspected small bowel bleeding once the upper GI tract and colon are satisfactorily cleared by EGD/PE and colonoscopy. VCE allows for direct visualization of the entire small bowel mucosa. In a meta-analysis of 14 prospective studies investigating small bowel bleeding, VCE had a higher diagnostic yield than PE (56% *vs.* 26%), or small bowel follow through (6%) [130]. VCE is also considered superior to IOE, CTA, and standard angiography [131,132]. The overall diagnostic yield of VCE is 35–77%; the yield is higher among those who are inpatient, earlier VCE (within 2 weeks, greatest yield between 48 to 72 h of a bleeding episode), overt bleeding with transfusion requirement, Hgb <10 g/dL, longer duration of bleeding (>6 months), male sex, increasing age, liver, cardiac and renal comorbidities and use of anticoagulation [9,127,133]. Studies indicate that findings on VCE have led to endoscopic or surgical intervention or change in medical management in 37–87% of patients, and in 50–66% of cases there is no rebleeding following VCE-directed interventions [134,135]. The rate of rebleeding in those with negative VCE is low, between 6–11% [136,137] as the negative predictive value (NPV) of VCE is 83–100% [135]. If the first VCE is negative, a second VCE may be beneficial and increase diagnostic yield, particularly when occult bleeding changes to overt or there is a decrease in Hgb ≥ 4 g/dL [138,139,140,141].

The main limitations of VCE use are inability to provide therapeutic intervention, and it may not detect clinically important lesions in the duodenum (because of rapid transit through the duodenal loop) and proximal jejunum [142,143]. Complications associated with VCE include a 2% risk of capsule retention in patients with surgical anastamosis or unknown strictures and rarely, risk of perforation [135,144].

### 3.3. Device Assisted Enteroscopy

Deep enteroscopy of the small bowel is the technique of choice for evaluation and therapeutic intervention of the mid-gut between the ampulla of Vater and the ileo-cecal valve [145], and is accomplished via DAE using double balloon enteroscopy (DBE), single balloon enteroscopy (SBE), and spiral enteroscopy (SE) [146]. The technique for DAE is based on different designs of an overtube which fits over a thin, flexible enteroscope designed to minimize looping of the small bowel while pleating it back over the overtube and the enteroscope.

DBE utilizes a “push and pull technique” and can be performed in an antegrade or retrograde fashion. The depth of intubation with DBE ranges from 240 cm to 260 cm past the ligament of Treitz via the antegrade approach and from 102 cm to 140 cm past the ileo-cecal valve through the retrograde approach [146]. The diagnostic yield of DBE for suspected small bowel bleeding and other small bowel disorders ranges from 60–80%, and success with endoscopic therapeutic interventions is estimated to be 40–73% [127]. In a systematic review [147] of 12,823 DBE procedures, the diagnostic yield of DBE was 68.1%, with vascular lesions (66%) being the most common finding; the rate of total enteroscopy i.e., intubation of the entire small bowel was 44%. The overall adverse event rate (pancreatitis, ileus, perforation, bleeding, aspiration pneumonia) associated with DBE is approximately 1%, going up to 3–4% in therapeutic DBE, and overall mortality is 0.05% [146]. In several meta-analyses the diagnostic yield of VCE and DBE was comparable [148,149,150,151]. The diagnostic yield of DBE may be higher in patients with a positive VCE [149] and the use of VCE guided DBE may increase the diagnostic and therapeutic yield of DBE [152]. However, in situations where there is a high index of suspicion for a small bowel lesion, DBE should be considered even when the VCE is negative [148]. Limited data from two studies suggests that early DBE alone in suspected small bowel bleeding may yield better outcomes [153,154].

SBE also utilizes a “push and pull technique” and can be performed in an antegrade or retrograde fashion. The depth of intubation with SBE ranges from 133 cm to 256 cm past the ligament of Treitz via the antegrade approach, and from 73 cm to 163 cm past the ileo-cecal valve through the retrograde approach. The rate of total enteroscopy is between 15–25%, and the diagnostic yield in suspected small bowel bleeding varies from 33–74% [127]. The overall adverse event rate is 1% [146]. The potential advantages of SBE over DBE include a shorter set up time, requirement for one balloon instead of two, an easier balloon control panel, and use of a non-latex balloon [155]. However, the two techniques are considered comparable for diagnostic and therapeutic evaluation of small bowel bleeding sources [156,157,158].

SE utilizes a rotational endoscopy technique to advance through the small bowel and is largely used via an antegrade approach. The retrograde approach was described in a pilot study of six patients [159]. The mean depth of intubation ranges from 176 cm to 250 cm. Although SE is a simpler procedure with a has a reduced time, it has a low rate of complete enteroscopy because of limited retrograde use. In a prospective study, the diagnostic yield in patients with a positive VCE was estimated to be 57–62% [160]. SE is also associated with improved outcomes with regards to transfusion requirements, iron supplementation, and additional therapeutic procedures [160,161]. Adverse events include minor mucosal tears and perforation in 0.3% of cases [146]. In a small trial that compared antegrade DBE in patients with suspected small bowel vascular lesions to SE, SE had a shorter insertion and examination time, but the depth of insertion was greater with DBE [162]. More recent prospective data indicates that DBE and SE have a similar insertion time and distance, along with similar diagnostic and therapeutic yield [163].

### 3.4. Radiographic Imaging

Small bowel follow through and enteroclysis although previously recommended are no longer used in the routine evaluation of suspected small bowel bleeding [9,127]. Multiphase CTE can detect small bowel vascular, mucosal lesions and neoplasms, and can aid in decision making of an antegrade or retrograde approach in DAE [9,127]. VCE and CTE are complementary exams, although CTE may have an advantage over VCE in improved detection of small bowel masses [164]. There is limited data supporting the role of MR enterography in suspected small bowel bleeding [165,166].

Radioisotope bleeding scans can be useful to detect overt small bowel bleeding provided that the rate of bleeding is at least 0.1 to 0.4 mL/min [167]. The use of 99m Tc-labeled RBC scintigraphy is sensitive in detecting bleeding, but there is variability in localization of bleeding and need for further evaluation with angiography [168]. In young patients, a 99m Tc-pertechnetatae scan is useful for detection of Meckel’s diverticulum [169], and has a sensitivity of 62–87.5% for ectopic gastric mucosa [170,171].

Angiography may be useful in cases of small bowel bleeding where there is active bleeding at a rate of 0.5–1 mL/min, and also allows for transarterial embolization if a bleeding source is embolized. The yield for angiography is 30–77% [172,173,174].

## 4. Lower Gastrointestinal Bleeding

LGIB was historically defined as GI bleeding that originates from a site distal to the ligament of Treitz [175], but is now referred to as bleeding from a source distal to the ileocecal valve i.e., from the colon and rectum [176,177]. The annual incidence of LGIB is 0.03% and rises steadily with age [126,178,179]. The incidence of hospitalizations for LGIB is comparable to that of UGIB, largely because of a decrease in incidence of UGIB [3,4,180].

The most common etiologies of LGIB that are amenable to endoscopic intervention are diverticular bleeding, angioectasias, and post-polypectomy bleeding [181]. Other etiologies include ischemic or infectious colitis, hemorrhoids, IBD, NSAID colopathy, radiation proctitis, stercoral ulcers, Dieulafoy lesions, colorectal neoplasias, and unknown causes in 6–23% of cases [182]. A suggested algorithm for management of severe LGIB by the ASGE is shown in Figure 5 [176]. While colonoscopy is indicated as the examination of choice for diagnosis and treatment of acute LGIB, a meta-analysis of four RCTs found that early colonoscopy (within 24 h) did not reduce further bleeding or mortality among patients [183].

### Role of Endoscopy in Lower Gastrointestinal Bleeding

Diverticulosis is the most common cause of LGIB and its prevalence increases from less than 20% at age 40, to 60% at age 60 [184]. Diverticular bleeding is arterial and is caused by trauma to the vasa recta at the neck or dome of the diverticulum resulting in weakness of the artery and rupture into the lumen. Risk factors for diverticular bleeding include aspirin and NSAID use, advanced age, physical inactivity, hypertension, hyperlipidemia, chronic renal failure, and ischemic heart disease [185,186,187,188]. It presents as painless hematochezia that is typically self-limited. Endoscopic therapy can be attempted if active bleeding, non-bleeding visible vessel or an adherent clot (Figure 6) can be localized to a particular diverticulum during colonoscopy [189], which is difficult because the diverticula are usually numerous and bleeding may be intermittent. The use of clips over thermal therapy is potentially advantageous in that it avoids the possible risk of transmural injury and perforation. Perforation with contact thermal coagulation in the right side of the colon is seen in 2.5% of patients [190]. In active bleeding, dilute epinephrine (0.5–2 mL) can be injected at the site of active bleeding or around a non-bleeding visible vessel [191], followed by placement of hemoclips either direct over the vessel or clot, or by opposing the walls of the diverticulum and closing the diverticular orifice itself thereby resulting in bleeding tamponade [192]. This approach is not associated with early rebleeding, but late rebleeding is seen in 18% to 22% of patients [191,192]. Colonoscopy more commonly detects diverticular bleeding originating in the left side of the colon, whereas bleeding from the right colon which is the more common source is usually detected by angiography [126,193,194,195]. Placement of a tattoo or clip adjacent to a bleeding diverticulum should be considered to aid future localization if recurrent bleeding occurs [176,177]. In those patients with severe hematochezia who cannot undergo bowel preparation for a colonoscopy or cannot be stabilized, mesenteric angiography and embolization is considered an effective treatment for diverticular bleeding [196]. Surgery, preferably a segmental colectomy is reserved for persistent or refractory diverticular bleeding [197,198,199].

Angioectasias (Figure 7) account for 3–30% of episodes of LGIB and are more common in the right side of the colon and the elderly [200,201,202,203,204,205,206]. Risk factors for development of colonic angioectasias are similar to that in other parts of the gut. Colonoscopy allows for diagnosis and therapeutic intervention. Contact and noncontact thermal coagulation can be used for treatment as described earlier. A full bowel preparation is required when performing APC in the colon due to risk of colonic gas explosion from inadequate preparation [207]. In addition, lower power settings are used to decrease risk of perforation in the right side of the colon.

Post-polypectomy bleeding accounts for 2–8% of acute LGIB [179,206]. Acute bleeding can occur immediately due to involvement of an underlying artery or inadequate coagulation of the polyp stalk, or can be delayed up to four weeks later which results from sloughing of the eschar that covers a blood vessel, or due to extension of necrosis to a submucosal vessel in the non-injured tissue following use of thermal modalities for polypectomy. Risk factors for post-polypectomy bleeding include polyp size ≥ 2 cm, right colon polyps, thick stalk and resumption of antithrombotic therapy [177]. Endoscopic hemostasis can be secured through clipping, or a thermal modality with or without the use of dilute epinephrine, band ligation [177]. Hemostatic nanopowder/spray can be used as salvage therapy [208].

Radiation proctopathy is defined as damage to the rectal epithelium due to radiation that is associated with none to minimal inflammation of the mucosa. Chronic radiation proctopathy, more so than acute radiation proctopathy, presents with rectal bleeding, usually develops 9–14 months post-radiation but can occur up to 30 years after exposure [209,210,211]. Endoscopic findings include numerous large telengectasias [212]. Endoscopic treatment includes APC, bipolar electrocoagulation, heater probe, and RFA [213]. APC in particular is effective in reducing short-term symptoms of chronic radiation proctopathy [214]. Hemospray may be considered as adjunctive therapy [215].

Rectal ulcers can present as severe hematochezia, and are usually seen in the setting of significant concurrent medical illnesses that necessitate intensive care unit admissions [216,217,218]. Endoscopic findings can range from active bleeding (50%) nonbleeding visible vessels (33%), adherent clots (17%) to clean based ulcers (82%) [216]. Endoscopic treatment is associated with early rebleeding in up to half of patients, and mortality rate may be as high as 48% among those who have high risk lesions and significant comorbidities [216,218].

Colorectal neoplasias in the left colon can result in hematochezia, whereas those on the right side typically present with chronic blood loss and IDA [219]. Bleeding occurs as a result of overlying ulceration or erosion into a vessel by the tumor. Endoscopic treatment is limited. Hemospray may be useful as a temporizing measure [220].

Persistent or recurrent bleeding following endoscopic intervention can be seen in 25% of cases of LGIB [221,222] and may require angiographic embolization [109,175]. However, in comparison to effective endoscopic treatment, embolization is associated with early rebleeding in 22% of cases of LGIB [206].

Colonoscopy is used for diagnostic evaluation of LGIB associated with ischemic colitis, IBD, and hemorrhoids but treatment is primarily medical management and severe cases that are refractory to medical therapy may require surgical intervention [176,177].

## 5. Future Direction

The use of artificial intelligence (AI) in the diagnosis and management of gastrointestinal diseases, including GIB is gaining momentum [223]. In twelve studies [224,225,226,227,228,229,230,231,232,233,234,235] investigating the use of AI in detection of small bowel bleeding, small bowel bleeding was identified with an accuracy of >90%. Machine learned AI models developed for management of UGIB and LGIB were able to identify patients at risk of recurrent bleeding with 90% accuracy [236,237,238,239,240]. In two RCTs [241,242] that compared the efficacy of endoscopy with or without AI based algorithms, the use of AI improved the diagnostic yield of endoscopic procedures. Further advances in the application of AI in management of GIB are anticipated and may improve both diagnosis, and endoscopic management of GIB in the future.

## 6. Conclusions

Non-variceal gastrointestinal hemorrhage is an important medical emergency. Prompt recognition, and timely institution of medical, and endoscopic hemostatic therapy, where applicable improves outcomes. As novel diagnostic and therapeutic endoscopic capabilities continue to evolve and gain applicability, further improvements in prognosis can be anticipated.

## Figures and Tables

**Figure 1 jcm-09-00402-f001:**
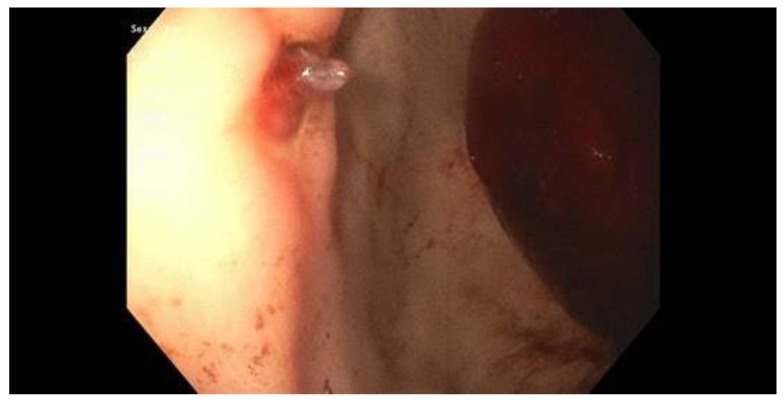
Gastric ulcer with a non-bleeding visible vessel (Forrest IIa ulcer) seen along the lesser curvature. A large, fresh clot is seen along the greater curvature extending into the antrum.

**Figure 2 jcm-09-00402-f002:**
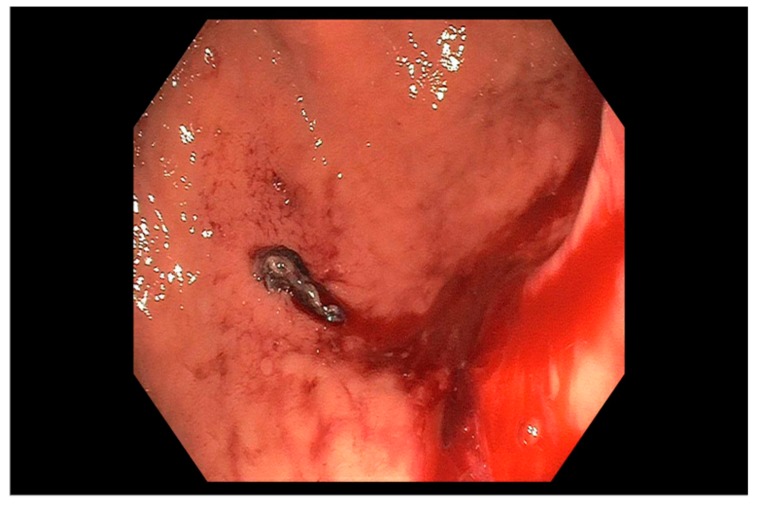
Gastric ulcer with an adherent clot (Forrest IIb ulcer) with active oozing seen in the gastric body.

**Figure 3 jcm-09-00402-f003:**
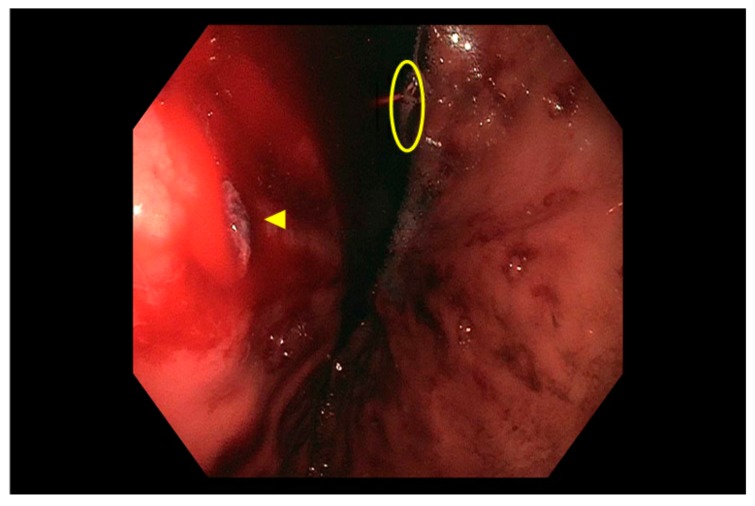
Actively bleeding Dieulafoy lesion seen in gastric fundus (indicated by arrow) upon retroflexion (position of gastroscope indicated by circle).

**Figure 4 jcm-09-00402-f004:**
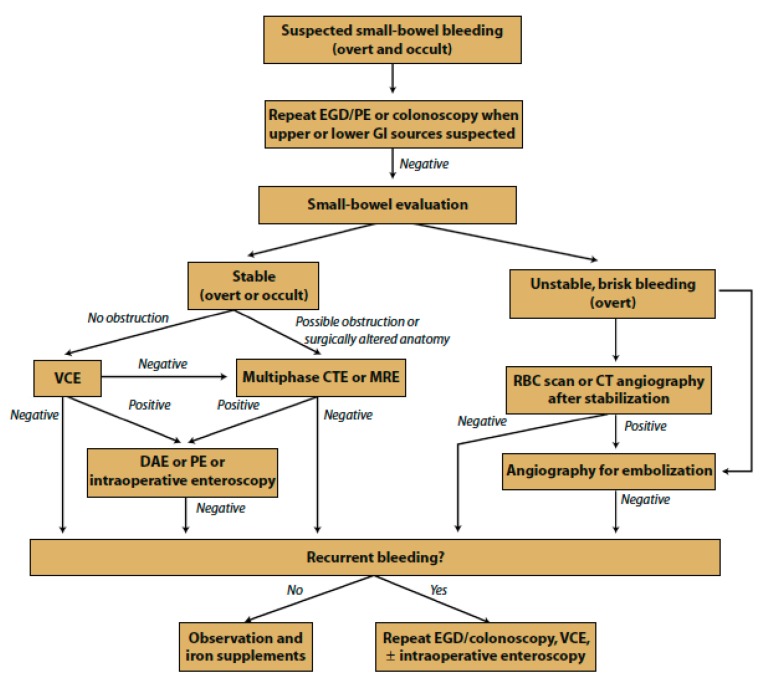
Suggested management approach to overt and occult small-bowel bleeding. *PE,* push enteroscopy; VCE, video capsule endoscopy; *DAE,* device-assisted enteroscopy; *CTE,* CT enterography; *MRE*, magnetic resonance enterography; *RBC,* red blood cell. Copyright © 2017 by the American Society for Gastrointestinal Endoscopy 0016-5107. Reprinted with permission from Gurudu et. al., The role of endoscopy in management of suspected small-bowel bleeding, GIE, 2017, Volume 85, Issue 1, Pages 22–31.

**Figure 5 jcm-09-00402-f005:**
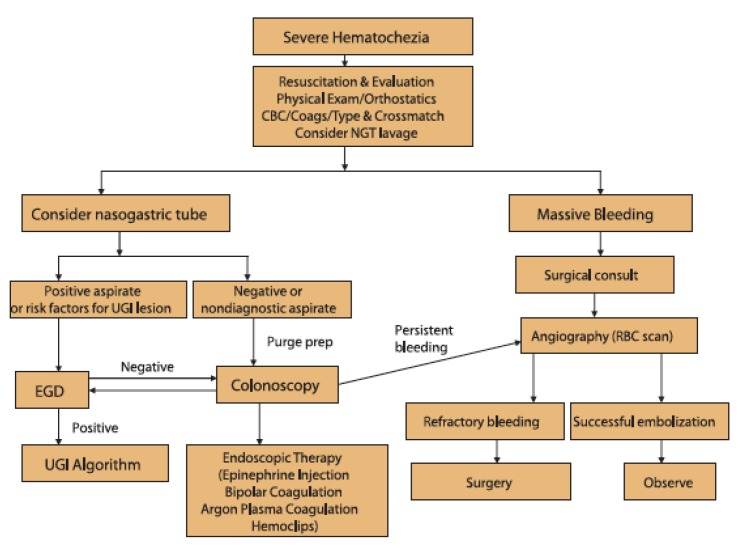
Management of severe hematochezia. Copyright © 2014 by the American Society for Gastrointestinal Endoscopy 0016-5107. Reprinted with permission from Pasha et. al., The role of endoscopy in the patient with lower GI bleeding, GIE, 2014, Volume 79, Issue 6, Pages 875-885.

**Figure 6 jcm-09-00402-f006:**
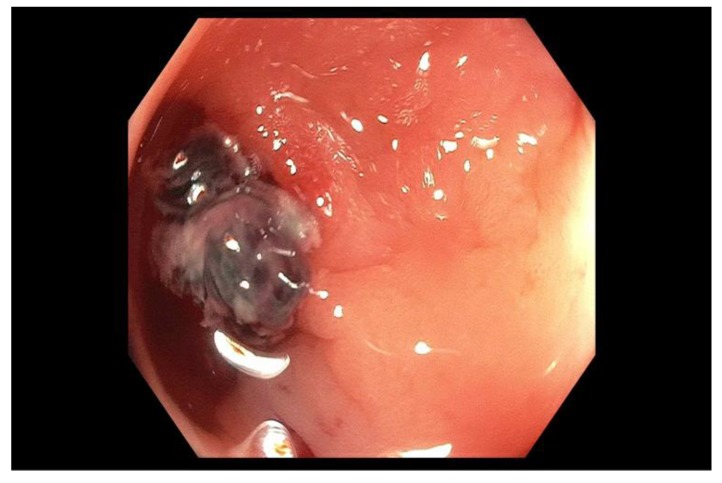
Adherent clot with oozing blood seen within a diverticulum in the ascending colon.

**Figure 7 jcm-09-00402-f007:**
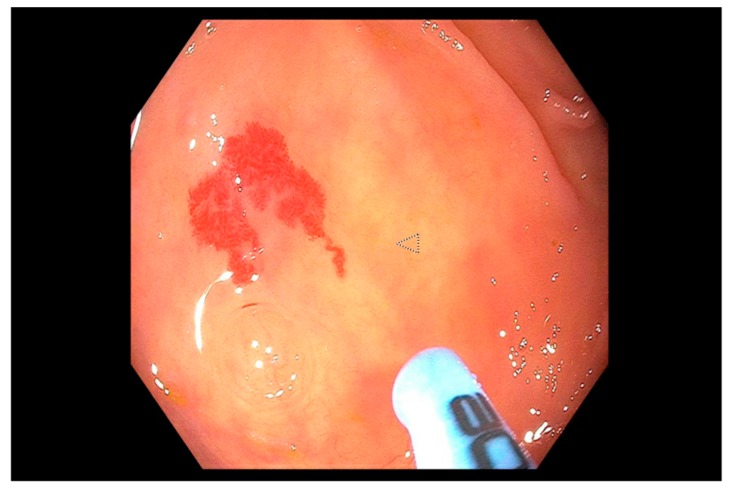
Two large arteriovenous malformations (AVMs) seen in the ascending colon (indicated by arrow) which were treated with argon plasma coagulation (APC).

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
