# Peer review of "Diagnosis and Management of Non-Variceal Gastrointestinal Hemorrhage: A Review of Current Guidelines and Future Perspectives"

_jcm, 2020, doi:10.3390/jcm9020402_

Round 1
Reviewer 1 Report
Well-written review of both diagnosis and management of GI hemorrhage. I appreciate the author's fluid writing style, key references shared, and high quality images to support the text.
Author Response
Response to Reviewer 1 comments:
Thank you for your review of our manuscript. Your feedback is greatly appreciated.
Reviewer 2 Report
The authors present a review of the literature focusing on the investigation and management of non-variceal gastrointestinal hemorrhage.
While this is a topic that has been extensively covered previously in the literature, the focus of this review is on management, in particular, that of endoscopic management and treatment selection based on diagnostic findings.
The review is simple, succinct and references major studies which have contributed to practice changes.
However, one of the objectives outlined in the abstract includes providing insight in to future directions. I do not feel that this was adequately addressed or covered in the manuscript.
The statement ‘coffee-ground emesis suggests more limited bleeding that stopped sometime ago’ is incorrect as coffee-ground emesis, particularly if multiple and persisting with hemodynamic compromise can suggest significant active upper gastrointestinal hemorrhage.
The section on medical management also needs to address the importance of stopping antiplatelet and anticoagulant therapy and correcting coagulopathy if present.
In the section of role for endoscopy in lower gastrointestinal bleeding the risk of angioembolisation in the management of colonic lesions is significant and an important consideration in treatment selection as attempts should be made to ensure adequate bowel preparation and endoscopic management prior to consideration of angioembolisation.
This review is a simple and broad description of the diagnosis and management of upper, lower and occult gastrointestinal bleeding.
Author Response
Dear Reviewer 3,
Thank you for your feedback, your review is greatly appreciated. Please see our answers below to your queries.
Point 1: However, one of the objectives outlined in the abstract includes providing insight in to future directions. I do not feel that this was adequately addressed or covered in the manuscript.
Response 1: We have added on a section titled "Future Directions" which follows the section on lower gastrointestinal bleeding. In that section we have included the possible role of artificial intelligence in the management of gastrointestinal bleeding.
Point 2: The statement ‘coffee-ground emesis suggests more limited bleeding that stopped sometime ago’ is incorrect as coffee-ground emesis, particularly if multiple and persisting with hemodynamic compromise can suggest significant active upper gastrointestinal hemorrhage.
Response 2: We have removed this statement from the manuscript in the section on "Bleeding manifestations".
Point 3: The section on medical management also needs to address the importance of stopping antiplatelet and anticoagulant therapy and correcting coagulopathy if present.
Response 3: We have added to the section on initial assessment about anti thrombotic agents and have included a reference to the guidelines published by the American Society for Gastrointestinal Endoscopy.
Point 4: In the section of role for endoscopy in lower gastrointestinal bleeding the risk of angioembolisation in the management of colonic lesions is significant and an important consideration in treatment selection as attempts should be made to ensure adequate bowel preparation and endoscopic management prior to consideration of angioembolisation.
Response 4: The consideration of embolization in the section in lower gastrointestinal bleeding that is not amenable or refractory to endoscopic management is mentioned in the section on lower gastrointestinal bleeding. We did not elaborate on it as we wanted to keep the focus of our paper on endoscopic management.
Reviewer 3 Report
In this manuscript by Mujtaba et al., the authors present a briefly reviewed on the diagnosis and management of non-variceal gastrointestinal hemorrhage. They focused on etiologies, manifestations and recommended guidelines on the management of gastrointestinal bleedings, particularly in the use of endoscopy diagnosis.
This review is very interesting not only for experts in GI but for a broad spectra of readers. It well written, and very easy to followed. The description of different etiologies, diagnosis, devices to use risks, comparison of treatments in “chapters” for each portion of the GI strengthen the work.
This work fits the aims and scope of the journal, provides relevant knowledge in the field through a clear review.
Author Response
Response to Reviewer 3 comments:
Thank you for your review of our manuscript. Your feedback is greatly appreciated.